# The Desmin Mutation *DES*-c.735G>C Causes Severe Restrictive Cardiomyopathy by Inducing In-Frame Skipping of Exon-3

**DOI:** 10.3390/biomedicines9101400

**Published:** 2021-10-05

**Authors:** Andreas Brodehl, Carsten Hain, Franziska Flottmann, Sandra Ratnavadivel, Anna Gaertner, Bärbel Klauke, Jörn Kalinowski, Hermann Körperich, Jan Gummert, Lech Paluszkiewicz, Marcus-André Deutsch, Hendrik Milting

**Affiliations:** 1Heart and Diabetes Center NRW, Erich and Hanna Klessmann Institute, University Hospital of the Ruhr-University Bochum, Georgstrasse 11, D-32545 Bad Oeynhausen, Germany; franziska.flottmann@uni-bielefeld.de (F.F.); sratnavadivel@hdz-nrw.de (S.R.); agaertner@hdz-nrw.de (A.G.); bklauke@hdz-nrw.de (B.K.); jgummert@hdz-nrw.de (J.G.); 2Microbial Genomics and Biotechnology, Center for Biotechnology, Bielefeld University, D-33615 Bielefeld, Germany; chain@cebitec.uni-bielefeld.de (C.H.); joern@CeBiTec.Uni-Bielefeld.de (J.K.); 3Clinic for General and Interventional Cardiology/Angiology, Heart and Diabetes Center NRW, University Hospital of the Ruhr-University Bochum, Georgstrasse 11, D-32545 Bad Oeynhausen, Germany; 4Heart and Diabetes Center NRW, Institute for Radiology, Nuclear Medicine and Molecular Imaging, University Hospital of the Ruhr-University Bochum, Georgstrasse 11, D-32545 Bad Oeynhausen, Germany; hkoerperich@hdz-nrw.de; 5Heart and Diabetes Center NRW, Department of Thoracic and Cardiovascular Surgery, University Hospital Ruhr-University Bochum, Georgstrasse 11, D-32545 Bad Oeynhausen, Germany; lpaluszkiewicz@hdz-nrw.de (L.P.); mdeutsch@hdz-nrw.de (M.-A.D.)

**Keywords:** restrictive cardiomyopathy, skeletal myopathy, desmin, intermediate filaments, desmosomes, cardiovascular genetics

## Abstract

Currently, little is known about the genetic background of restrictive cardiomyopathy (RCM). Herein, we screened an index patient with RCM in combination with atrial fibrillation using a next generation sequencing (NGS) approach and identified the heterozygous mutation *DES*-c.735G>C. As *DES*-c.735G>C affects the last base pair of exon-3, it is unknown whether putative missense or splice site mutations are caused. Therefore, we applied nanopore amplicon sequencing revealing the expression of a transcript without exon-3 in the explanted myocardial tissue of the index patient. Western blot analysis verified this finding at the protein level. In addition, we performed cell culture experiments revealing an abnormal cytoplasmic aggregation of the truncated desmin form (p.D214-E245del) but not of the missense variant (p.E245D). In conclusion, we show that *DES*-c.735G>C causes a splicing defect leading to exon-3 skipping of the *DES* gene. *DES*-c.735G>C can be classified as a pathogenic mutation associated with RCM and atrial fibrillation. In the future, this finding might have relevance for the genetic understanding of similar cases.

## 1. Introduction

Desmin, encoded by the *DES* gene, is the major specific intermediate filament (IF) protein. Mutations in *DES* cause different cardiac and skeletal myopathies [1,2] or combinations of both [3]. Although the exact incidence of pathogenic *DES* mutations is unknown, desminopathy is a rare disease with an estimated incidence of less than 1 in 2000 [4]. Desmin consists of an α-helical rod domain flanked by non-helical head and tail domains. It forms coiled-coil dimers, which anneal antiparallel into tetramers [5]. Eight antiparallel tetramers form unit-length filaments (ULFs), which are the essential building blocks of intermediate filaments [4]. 

Desmin filaments connect different cell organelles and multi-protein complexes, like the cardiac desmosomes, costameres, and Z-bands, and are, therefore, highly relevant for the structural integrity of cardiomyocytes [6]. The majority of known pathogenic *DES* mutations are missense mutations or small in-frame deletions that potentially change the physical properties of desmin [4,7,8]. Since prolines destabilize α-helices, many pathogenic *DES* missense mutations leading to an exchange against proline have been described [9]. *DES* mutations interfere at different stages within the filament assembly process leading to an abnormal cytoplasmic desmin aggregation [10]. Heterozygous splice site mutations or other loss of function mutations in the *DES* gene are rare [11,12].

Herein, we describe an index patient with a heterozygous in-frame exon skipping desminopathy who developed severe restrictive cardiomyopathy (RCM) in combination with atrial fibrillation and, finally, underwent heart transplantation (HTx). The majority of RCM associated mutations have been described in genes encoding sarcomeric proteins, like cardiac troponins or filamin-C [13,14,15,16,17]. Since several different genes are associated with RCM, we performed NGS analysis revealing the heterozygous *DES*-c.735G>C mutation, which is most likely disease causing within the described family. Several other family members were affected by skeletal or cardiac myopathies. *DES*-c.735G>C might cause the exchange of glutamate against aspartate at position 245 (p.E245D). 

However, the mutant nucleotide is the last one of exon-3. Previously, Clemen et al. demonstrated in skeletal muscle tissue that in addition to the missense exchange (p.E245D) an exon skipping is induced by this mutation [18]. This exon skipping leads to an in-frame deletion of 96 base pairs (32 amino acids). However, the ratio of the missense and the deletion mutations in the human heart remains unknown. Therefore, we investigated by nanopore sequencing the myocardial expression levels of mutant and wild-type *DES* transcripts. Of note, these experiments revealed skipping of the *DES* exon-3 but excluded p.E245D transcripts. 

Additionally, we generated expression constructs of the missense mutation and of the in-frame deletion (p.D214-E245del) resulting from exon-3 skipping and analysed the filament assembly in cell culture in combination with confocal microscopy revealing an abnormal cytoplasmic aggregation of the in-frame exon deletion but not of the missense mutation as previously described for several other *DES* mutations [19,20,21]. Immunohistochemistry (IHC) confirmed likewise desmin aggregates and degraded sarcomeres in the explanted myocardial tissue of the index patient.

In conclusion, we demonstrated by nanopore sequencing that an in-frame exon skipping is caused by *DES*-c.735G>C leading to a filament assembly defect of the mutant desmin, which is likely causative for RCM.

## 2. Materials and Methods

### 2.1. Clinical Description of the Index Patient (III-9)

The index patient presented decompensated right heart failure at the age of 41 years and was admitted with edema of the legs, hepatomegaly, shortness of breath (NYHA III), nycturia, and palpitations. Electrocardiogram (ECG) analyses revealed atrial fibrillation. Transthoracic echocardiography (TTE) analyses revealed moderate to severe tricuspid valve regurgitation and massive dilation of the right atrium (RA) with associated spontaneous echo contrast. Slight dilation of the right ventricle (RV) but excluded left-ventricular (LV) dilation (Figure 1A,B). 

While systolic left-ventricular ejection fraction (LVEF) was preserved mitral inflow signal analysis revealed severe diastolic dysfunction. In addition, a wall-adherent thrombus in the RA was diagnosed. Cardiac magnetic resonance imaging (MRI) verified intense dilation of the RA (end-diastolic area about 60 cm^2^) and moderate dilation of the LA (end-diastolic area 34 cm^2^) (Figure 1C–E and Appendix A). 

The LV diameters were normal (LV-EDD 39 mm and LV-ESD 26 mm) and the RV diameters were slightly increased (RV-EDD 35 mm and RV-ESD 22 mm RV myocardial biopsies revealed an increased number (>7 cells/mm^2^) of activated T-cells (CD45R0) and macrophages (CD68) indicating myocardial inflammation (Figure 1F,G) [22]. Due to progressive clinical worsening (Ergospirometry: VO_2_max 9.81 mL/kgKG/min; right-heart catheterization (20 h after levosimendan therapy): PCWP 15 mmHg, CI 1.4 L/min/m^2^), the patient was listed for highly urgent HTx). He finally underwent orthotopic HTx at the age of 43. In total, the clinical presentation of III-9 is in good agreement with the diagnosis of RCM.

### 2.2. Genetic Analyses

The family anamnesis of the index patient (III-9, Figure 2) revealed five further family members with skeletal and/or cardiac myopathies. His father (II-5) was deceased by an unclassified cardiomyopathy, and two uncles (II-1 and II-3) and a cousin (III-5) developed skeletal myopathy. Of note, II-1 additionally developed cardiomyopathy and underwent HTx. Furthermore, the grandmother (I-2) developed an unspecified cardiomyopathy. Detailed clinical data of the affected family members were not available. 

After identification of significant family history of skeletal and cardiac myopathies, we applied the TrueSight Cardio NGS panel (Illumina, San Diego, CA, USA) covering the most likely cardiomyopathy associated genes (see the Appendix B for a complete gene list) to investigate the putative underlying mutations in the index patient (III-9). The MiSeq system (Illumina) was used for NGS. No genomic DNA was available from further family members to perform co-segregation analysis within the family. A minor allele frequency (MAF) < 0.001 was used for filtering of identified sequence variants. Sanger sequencing was used to verify *DES*-c.735G>C using appropriate primers (Table 1).

### 2.3. Reverse Transcription Polymerase Chain Reaction

The total RNA was extracted from about 30 mg myocardial tissue from the index patient (III-9) and a rejected donor heart (non-failing, NF) using the RNeasy Mini Kit (Qiagen, Hilden, Germany) according to the manufacturer’s instructions. We transcribed 1.2 µg total RNA into cDNA using SuperScript II reverse transcriptase (Thermo Fisher Scientific, Waltham, MA, USA) in combination with oligo(dT)_18_ primers (Table 1) according to the manufacturer’s instructions. Reverse transcription polymerase chain reaction (RT-PCR) was performed using the appropriate primers (Table 1, 1 µM), Phusion DNA polymerase, and HF buffer (Thermo Fisher Scientific). The annealing temperature was 60 °C, and 35 cycles were used for PCR amplification. The full-length PCR products were purified with the GeneJET Gel Extraction Kit (Thermo Fisher Scientific) and were processed to nanopore sequencing.

### 2.4. Amplicon Nanopore Sequencing

*DES* cDNA was sequenced using the SQK-LSK109 kit on a GridION with 9.4.1. flow-cells (Oxford Nanopore Technologies, Cambridge, UK). Base calling was carried out with guppy v5.0.11 and the super-accurate base call model. Fastq data was adapter trimmed using porechop v0.2.4 (https://github.com/rrwick/Porechop, accessed on 28 July 2021) and mapped on the human reference genome hg38 using minimap2 v2.10-r761 with the -x splice parameter [23]. 

Alignment sorting and bam conversion was carried out using samtools v1.11. Isoform analysis was carried out using FLAIR v1.5.1. with the align, correct, collapse, and quantify steps [24]. Isoforms with less than 1% of reads supported were discarded.

### 2.5. Immunohistochemistry

Explanted septal, left-, and right–ventricular myocardial tissue was fixed in 4% Roti Histofix (Carl Roth, Karlsruhe, Germany) and was embedded in paraffin. We prepared 5 µm sections using a microtome (Leica, Wetzlar, Germany) that were deparaffinized using xylene and ethanol as described [25]. Bovine serum albumin (5% in phosphate buffered saline, PBS) was used for blocking (30 min, room temperature). Polyclonal goat anti-desmin antibodies (15 µg/mL, #AF3844, R&D Systems, Minneapolis, MN, USA) were used in combination with secondary anti-goat antibodies conjugated to Cy3 (1:100, #C2821, Sigma-Aldrich, St. Louis, MO, USA) for desmin labelling. We used 4′,6-diamidino-2-phenylindole (DAPI, 1 µg/mL) for nuclei staining (5 min, RT). Myocardial tissue was embedded using Fluorescent Mounting Medium (Dako, Glostrup, Denmark). Confocal microscopy was performed as previously described [26].

### 2.6. Plasmid Generation

The plasmid pEYFP-N1-*DES* was previously described [27]. The QuikChange Lightning Site-Directed Mutagenesis (SDM) Kit was used according to the manufacturer’s instruction to insert the missense variant *DES*-p.E245D and the deletion *DES*-p.D214-E245del into this plasmid using appropriate oligonucleotides (Table 1). The *DES* encoding sequences of all three plasmids were verified using Sanger sequencing (Macrogen, Amsterdam, The Netherlands). For details, see the Appendix A.

### 2.7. Cell Culture and Confocal Microscopy

The cell line SW13 does not express any cytoplasmic IF proteins and is, therefore, frequently used to investigate the effects of *DES* mutations [28]. SW13 cells were cultured in Dulbecco’s Modified Eagle’s Medium (DMEM) supplemented with 10% fetal calf serum and penicilline/streptomycine under standard conditions (37 °C, 5% CO_2_). Cells were cultured in µSlide chambers (ibidi, Martinsried, Germany) and were transfected using Lipofectamin 3000 according to the manufacturer’s instruction (Thermo Fisher Scientific). After 24 h of transfection, the cells were washed with PBS and fixed for 10 min with 4% Roti Histofix (Carl Roth, Karlsruhe, Germany) at RT. 

Afterwards, the cells were washed gently with PBS and were incubated with 0.1% Triton-X-100 for 15 min at RT. Phalloidin conjugated with Texas-Red-X (1:40, # T7471, Thermo Fisher Scientific) and DAPI (1 µg/mL) were used for the costaining of F-actin and the nuclei. Confocal microscopy was performed as described [29]. Approximately 100 cells were analyzed in each transfection experiment (*n* = 4).

### 2.8. Western Blot Analysis

About 50 mg left-ventricular myocardial tissue from a control sample (NF) and the index patient III-9 were homogenized and lysed in RIPA lysis buffer [30] supplemented with proteinase inhibitors. Protein concentrations were determined using the Pierce 660 nm Protein Assay (Thermo Fisher Scientific) in combination with the Infinite M1000 plate reader (Tecan, Männedorf, Switzerland). Western blot analysis was performed using chemiluminescence measurement as previously described [27].

### 2.9. Statistical Analysis

About 100 cells per independent transfection experiment (*n* = 4) were analyzed by counting the percentage of aggregate forming cells. A non-parametric Mann–Whitney test was used for analysis using GraphPad Prism 8.3 (GraphPad Software, San Diego, CA, USA). *p*-values ≤ 0.05 were considered as significant.

## 3. Results

The index patient (III-9, Figure 2) developed severe RCM and received HTx at the age of 43. Family anamnesis revealed five further family members (I-2, II-1, II-3, II-5, and III-5, Figure 1) affected by cardiomyopathy and/or skeletal myopathy indicating an autosomal-dominant mode of inheritance.

We performed a genetic analysis using a broad NGS gene panel revealing heterozygous *DES*-c.735G>C as the most likely pathogenic variant. The MAFs of all other identified variants were higher than the estimated prevalence of RCM. Interestingly, *DES*-c.735G>C changes the last base pair of *DES* exon-3 (Figure 3A). Sanger sequencing confirmed the presence of this *DES* mutation (Figure 3B).

Since the affected last base pair of exon-3 is part of a relatively conserved splice site, it is possible that this mutation causes a splicing defect (p.D214-E245del) and/or an amino acid exchange (p.E245D). To address this issue, we used RT-PCR in combination with nanopore sequencing to identify the myocardial *DES* transcripts in the index patient. In addition to the wild-type form, additional transcripts without the *DES* exon-3 were found in the patient sample but not in the non-failing control sample (Figure 4). Notably, we were unable to detect significant transcripts leading to the amino acid exchange p.E245D indicating that exon-3 skipping is the underlying pathomechanism.

To verify the results of the nanopore sequencing at the protein level, we performed western blotting (Figure 5). The skipping of exon-3 causes an in-frame deletion leading to a truncated protein (p.D214-E245del). Accordingly, we detected, in addition to the wild-type desmin (~55 kDa), a second smaller band (~50 kDa) using left-ventricular myocardial tissue from the index patient III-9 but not in case of the control sample (Figure 5).

In the next step, we introduced p.E245D and p.D214-E245del into expression plasmids (Appendix A) and performed transfection experiments in SW13 cells, which do not express any cytoplasmic intermediate filament proteins. Analyses using confocal microscopy revealed for wild-type and p.E245D intermediate filaments, whereas the expression of desmin-p.D214-E245del induced an abnormal cytoplasmic aggregation indicating a severe filament assembly defect (Figure 6A). Quantification revealed that all cells expressing mutant desmin-p.D214-E245del were positive for these cytoplasmic aggregates (Figure 6B).

In addition, we performed IHC analyses using explanted tissue from all three myocardial layers revealing sarcomeric structures with irregular desmin signals as well as desmin positive aggregates in the heart of the index patient (III-9). Notably, desmin staining at the intercalated disc was not observed in the myocardial tissue from III-9 (Figure 7).

## 4. Discussion

*DES* mutations cause a broad spectrum of myopathies and different cardiomyopathies [31], including RCM [1,32,33,34]. Most of these *DES* mutations cause single amino acid exchanges, which interfere with desmin filament assembly at different molecular stages [10,28]. In this study, using an NGS approach, we identified the desmin mutation *DES*-c.735G>C in an index patient from a family, in which several members developed skeletal myopathies or cardiomyopathies. 

Since we do not have gDNA from further family members, we were unable to perform a co-segregation analysis of *DES*-c.735G>C in the family. However, *DES*-c.735G>C is absent in the Genome Aggregation Database (gnomAD, https://gnomad.broadinstitute.org/, 6 August 2021) and in the NHLBI GO Exome Sequencing Project (https://evs.gs.washington.edu/, 6 August 2021). According to the guidelines of the American College of Medical Genetics and Genomics (ACMG) absence from controls is a moderate criterion for pathogenicity (PM2, ACMG guidelines) [35]. 

Notably, this mutation has been previously classified as a pathogenic variant found in patients with RCM in combination with atrial fibrillation [36], patients with distal myopathy in combination with cardiac conduction disease [18,37], or in patients with hypertrophic cardiomyopathy (HCM) in combination with cardiac conduction disease [38]. Classifying genetic mutations as ‘pathogenic’ in the literature without independent evaluation is a supportive criterion (PP5, ACMG guidelines). *DES*-c.735G>C is changing the last base pair of exon-3. Therefore, a damaging effect arising from a putative missense mutation (p.E245D) or a splice defect might be causative. 

Previously, Clemen et al. performed RT-PCR in combination with cloning and Sanger sequencing and revealed the expression of both mutant forms (p.E245D and p.D214-E245del) in the skeletal muscle of their patients [18]. However, whether the *DES*-c.735G>C mutation also leads to the expression of both mutant desmin species in the myocardial tissue is unknown. Therefore, we performed full length RT-PCR in combination with nanopore amplicon sequencing. 

As expected due to the heterozygous status of the index patient III-9, these experiments revealed the expression of the wild-type form as well as of *DES*-r.640-735del. However, transcripts encoding for *DES*-p.E245D have not been found in significant quantity. These experiments indicate that the truncated desmin caused by skipping of exon-3 is the pathogenic desmin species in the myocardial tissue. To verify these findings, we performed western blotting corroborating the expression of desmin-p.D214-E245del at the protein level. Changes in the protein length due to in-frame deletions are a moderate criterion for pathogenicity (PM4, ACMG guidelines) [35].

Most of the pathogenic *DES* mutations cause an abnormal cytoplasmic desmin aggregation [27,39]. Therefore, we inserted *DES*-p.D214-E245del and *DES*-p.E245D into expression plasmids and analyzed the filament assembly in transfected SW13 cells. These experiments revealed an abnormal cytoplasmic desmin aggregation of the truncated form but not of the missense mutation p.E245D when compared to wild-type desmin. Previously, Bär et al. showed that desmin-p.E245D forms regular intermediate filaments in transfected SW13 cells and does not interfere significantly with filament assembly using recombinant mutant desmin [10]. 

According to our cell culture experiments, we have found typical desmin-positive aggregates also in the explanted myocardial tissue of III-9. In general, functional studies are a strong criterion (PS3, ACMG guidelines) for pathogenicity according to the ACMG guidelines [35]. Therefore, *DES*-c.735G>C fulfills this criterion, as we have shown that the truncated desmin-p.D214-E245del causes an abnormal cytoplasmic aggregation as previously described for several other pathogenic *DES* mutations. In addition, this mutation is localized in the rod domain of desmin, which is a hot spot for pathogenic *DES* mutations [31], which is a moderate criterion (PM1, ACMG guidelines) for pathogenicity. Interestingly, several other pathogenic mutations affecting the donor splice-site of *DES* exon-3 have been previously described [40,41,42,43].

In summary, we have shown here that *DES*-c.735G>C causes a splicing defect in cardiac muscle leading to skipping of exon-3 and the expression of truncated desmin-p.D214-E245del, which is unable to form regular intermediate filaments in transfected cells. *DES*-c.735G>C fulfills one strong (PS3), one supportive (PP5), and three moderate criteria (PM1, PM2, and PM4) for its pathogenicity classification. In consequence, *DES*-c.735G>C has to be classified according to the ACMG guidelines as a pathogenic mutation associated with RCM.

## 5. Conclusions

By using an NGS approach, we identified the pathogenic *DES*-c.735G>C mutation in a patient with RCM. Using nanopore sequencing, we demonstrated that *DES*-c.735G>C causes a skipping of the third *DES* exon. Genetic and molecular analyses support pathogenicity of the caused splice site defect rather than a putative missense mutation p.E245D. In the future, our genetic and functional findings might be helpful for the genetic understanding of similar cases.

## Figures and Tables

**Figure 1 biomedicines-09-01400-f001:**
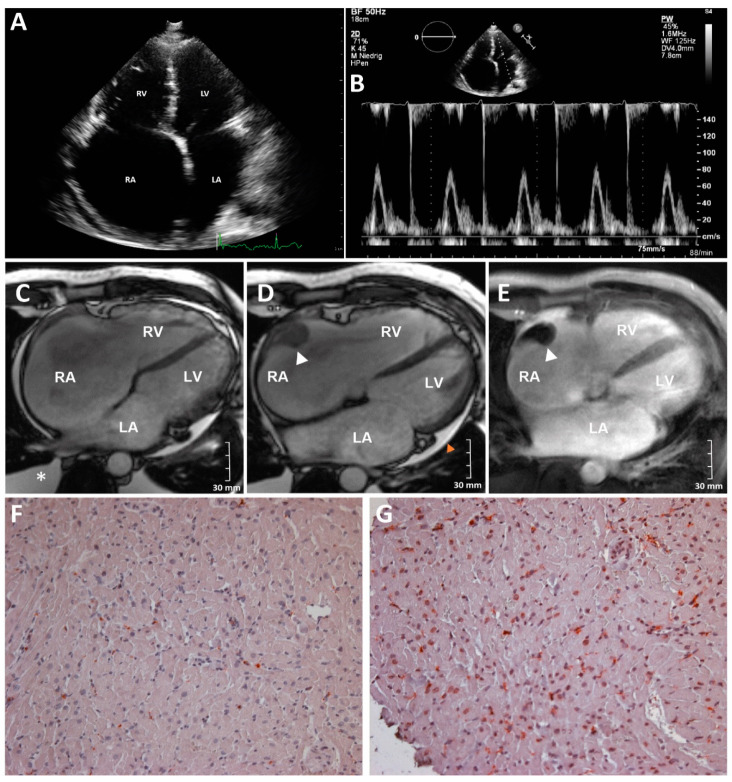
Clinical findings in index patient III-9 with RCM and persistent atrial fibrillation. (**A**) 2D transthoracic echocardiography. Apical four chamber view. Note enlargement of both atria with relatively small ventricles. A small amount of pericardial effusion is also visible. (**B**) Transthoracic echocardiography. Apical four chamber view, PW-Doppler of the mitral valve inflow. (**C**–**E**) Cardiac magnetic resonance imaging of III-9. (**C**,**D**) End-diastolic cine steady-state free-precession acquisitions. (**E**) Early 3D inversion-recovery T1-weighted fast gradient-echo for thrombus detection. (RA = right atrium; LA = left atrium; RV = right ventricle; and LV = left ventricle. A wall-adherent thrombus in the RA (34 × 25 × 17 mm) is marked with a white arrow head. Pericardial effusion (orange arrow head) was present, and pleural effusion (asterisk) was detected. (**F**,**G**) Immunohistology analysis of a right ventricular biopsy revealed myocardial inflammation. (200× magnification) (**F**) CD68 staining revealed increased number of macrophages. (**G**) CD45R0 staining revealed increased number of activated T-cells.

**Figure 2 biomedicines-09-01400-f002:**
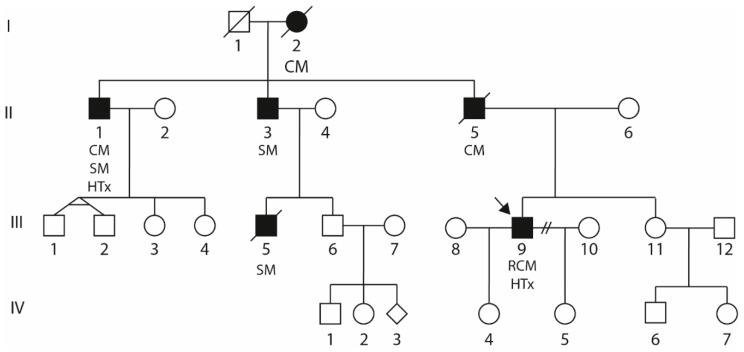
Pedigree of the described family. Circles represent females, squares represent males, and rhombs represent unknown gender. Black-filled symbols indicate a cardiac or skeletal muscle phenotype. Diagonal slashes indicate deceased people. The index patient (III-9) is marked with an arrow and carries *DES*-c.735G>C. CM = Cardiomyopathy; HTx = Heart transplantation; SM = Skeletal myopathy; and RCM = Restrictive cardiomyopathy.

**Figure 3 biomedicines-09-01400-f003:**
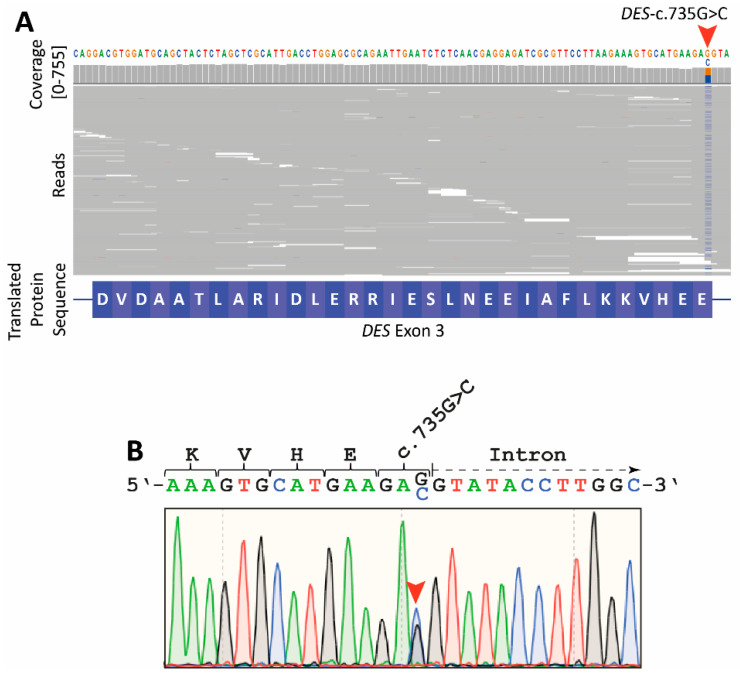
Genetic analysis of the index patient (III-9). (**A**) Integrated genome view of *DES* exon-3 revealed *DES*-c.735G>C in the gDNA from III-9 (red arrow). Cytosine was detected in 291 reads (53%, 131+, 160−), and guanosine was detected in 258 reads (47%, 119+. 139−). Reads are shown in grey. (**B**) Electropherogram of *DES*-c.735G>C generated by Sanger sequencing using gDNA from III-9 (red arrow). Of note, this missense mutation changes the last nucleotide in exon-3.

**Figure 4 biomedicines-09-01400-f004:**
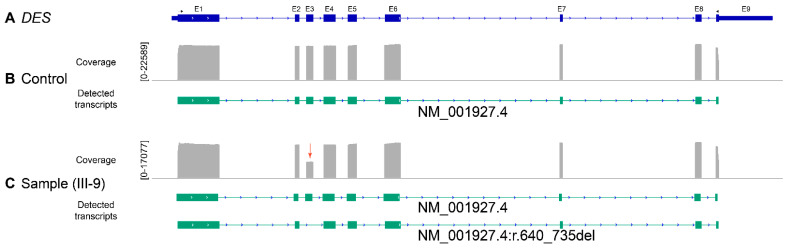
(**A**) Schematic overview of the *DES* gene. Arrow heads indicate the localization of the oligonucleotides used for amplification. Exons 1−9 are shown as blue boxes. Amplicon analysis of a control sample (**B**) and of the index patient III-9 (**C**). In addition to the regular transcript (NM_001927.4), a transcript missing exon-3 was found in III-9 (red arrow) but not in the control sample. Of note, transcripts encoding the missense variant p.E245D were absent in both samples indicating that *DES*-c.735G>C causes an in-frame exon skipping but not a missense variant in the myocardial tissue from the index patient III-9 (A:44; C:44; G:7523; T:8).

**Figure 5 biomedicines-09-01400-f005:**
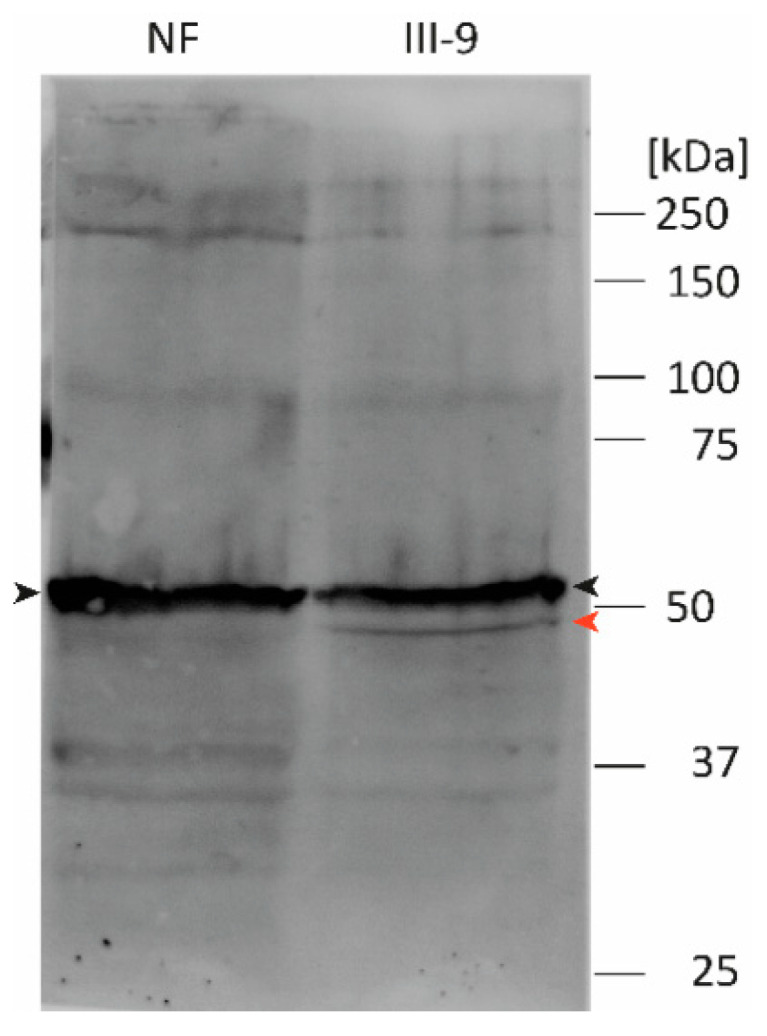
Western blotting analysis using myocardial tissue from a control sample (NF) and the index patient III-9. Of note, in the case of the index patient, two bands were observed, which correspond to wild-type desmin (black arrow head) and desmin-p.D214-E245del (red arrow head).

**Figure 6 biomedicines-09-01400-f006:**
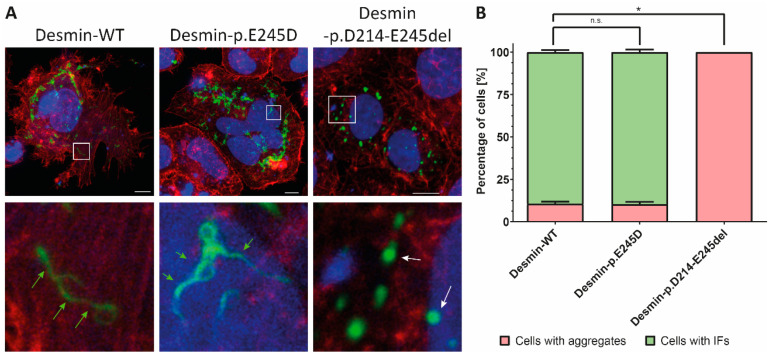
(**A**) Fluorescence analysis of transfected SW13 cells. Of note, desmin-p.D214-E245del causes severe desmin aggregation (white arrows), whereas wild-type and desmin-p.E245D form intermediate filaments (green arrows). Desmin is expressed as a fusion protein with EYFP (shown in green). F-actin is stained using phalloidin conjugated with Texas-Red (red) and nuclei are stained using DAPI (shown in blue). Scale bars represent 10 µm. (**B**) Quantification of desmin aggregate formation in transfected cells. About 100 cells have been analyzed per transfection experiment (*n* = 4). Error bars represent SD. Non-parametric Mann–Whitney-U-test was used for statistical analysis. n.s. = not significant; * indicate *p*-values < 0.05.

**Figure 7 biomedicines-09-01400-f007:**
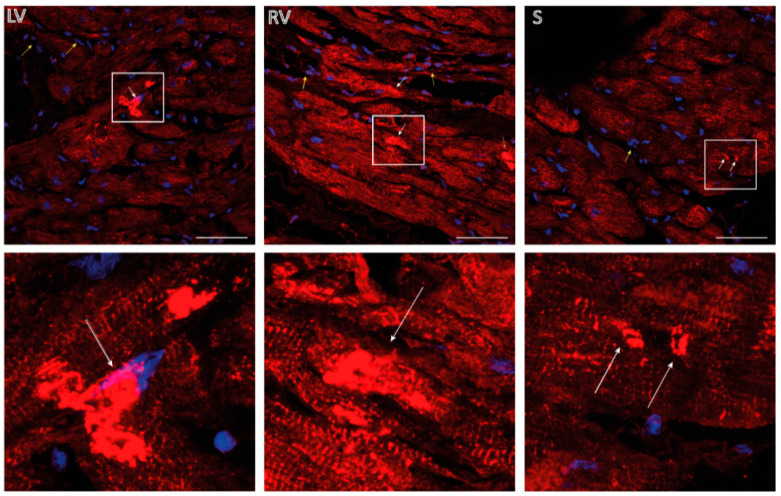
Immunohistochemistry analysis of explanted myocardial tissue from the index patient III-9. LV = left ventricular myocardial tissue; RV = right ventricular myocardial tissue; and S = septal myocardial tissue. Scale bars represent 50 µm. Desmin is shown in red. Nuclei were stained using DAPI and are shown in blue. Of note, desmin-positive aggregates were present in all three layers (white arrows). In addition, desmin-negative areas were present (yellow arrows) representing fibroblast or other non-cardiomyocyte cell types.

**Table 1 biomedicines-09-01400-t001:** Overview of the used oligonucleotides ^1^.

Name	Sequence (5′-3′)	Application
DES_3F	GGAAGAAGCAGAGAACAATTTGGC	Sanger sequencing
DES_3R	ACCTGGACCTGCTGTTCCTG	Sanger sequencing
oligo(dT)_18_	TTTTTTTTTTTTTTTTTT	Reverse transcription
DES_for	ATGAGCCAGGCCTACTCGTC	RT-PCR
DES_rev	GAGCACTTCATGCTGCTGCTG	RT-PCR
DES_E245D_for	TAAGAAAGTGCATGAAGACGAGATCCGTGAGTTGCAG	SDM
DES_E245D_rev	CTGCAACTCACGGATCTCGTCTTCATGCACTTTCTTA	SDM
DES_E3_Del_for	GCTGCCTTCCGAGCGGAGATCCGTGAGTTG	SDM
DES_E3_Del_rev	CAACTCACGGATCTCCGCTCGGAAGGCAGC	SDM

^1^ All oligonucleotides were purchased from Microsynth (Balgach, Switzerland). RT-PCR = reverse transcription polymerase chain reaction, and SDM = site directed mutagenesis.

## Data Availability

The data used and/or analyzed during the current study are available from the corresponding authors on reasonable request.

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
