# Peer review of "The Desmin Mutation DES-c.735G>C Causes Severe Restrictive Cardiomyopathy by Inducing In-Frame Skipping of Exon-3"

_biomedicines, 2021, doi:10.3390/biomedicines9101400_

Round 1

Reviewer 1 Report

Overall Review:

The work is highly focused on proving that mutation at the 3’ end of exon-3 is a direct or indirect cause of exon-3 skipping and thus production of difunctional protein causing Desmine protein aggregation and sickness of a patient.

Authors did a sequencing of a big panel of genes (TrueSight Cardio NGS panel) containing more than 100 genes connected to the cardiomyopathy. My question is were there other pathogenic mutations identified (in other genes) that could be also associated with observed patient symptoms?

If this was the case it should moderately weaken the overall conclusions. Patient have a point mutation in on average 50% of alleles (Illumina NGS on DNA) and based on Nanopore cDNA sequencing again ~50% (based on Fig 4) sequenced transcripts contain deletion of the whole exon-3 but only <1% sequences with the whole exon-3 had 735G>C mutation. This can be explained if we assume the expression seen in Nanopore data comes from only one allele. The <1% are “leaks” from the second allele and/or sequencing/base calling artifacts that are common for Nanopore technology (If I remember correctly 80-90% confidence in base calling of nucleotides?). So, it doesn’t convince me that there is connection between mutation and exon deletion. Are there any records of healthy people (without 735G>C mutation) with exon-3 lacking Desmine? Probably it will be hard to verify.

Authors presented figure with a Western blot DES detection analysis as a merge of 2-3 Western blots. Their “control” is showing “lack” of expression of the protein with exon-3 deletion but in my opinion the lower band is present also in the control (very week signal) and because the two protein samples were not run and exposed (detected with chemiluminescence?) together it is impossible to compare those signals. Also, we can see additional bands on a membrane so the antibody is probably detecting off-target proteins or dimmers?. Please provide pictures of full western blots in supplementary data.

According to supplementary data from Brodehl et al., 2012. [Dual Color Photoactivation Localization Microscopy of Cardiomyopathy-associated Desmin Mutants. 10.1074/jbc.M111.313841] Endogenus Desmin may come in two different forms ~54kDA and ~50kDA. Different isoforms are present in few cell lines. Base on that paper I’m assuming, the smaller band shouldn’t necessarily be counted as a unusual and pathogenic form?

Also (maybe it is my lack of knowledge) how often it happens that missense mutations in exons delete the whole exon (I understand that it happens with nonsense mutations but with just amino acid substitution?)?

I found this paper: Liu H.X., et al 2001. [A mechanism for exon skipping caused by nonsense or missense mutations in BRCA1 and other genes. DOI: 10.1038/83762] proving that it may happen at least in 5’ exon end but how it relates to Authors case with mutation at the 3’ of an exon. Is this paper supporting Authors findings?

I’m not fully convinced that mutation (not at the beginning of an exon) can cause whole exon deletion. Maybe the conclusion should be watered down a little to leave space for other interpretations? I don’t expect you to answers all of my questions but please do your best to convince me and future readers.

Very interesting work.

Some small details:

Material and methods

First line: ”… right heart failure at the age of 41…” there are two spaces between “failure” and “at”.

„…diastolic dysfunction. . In addition,…” the are two dots at the end of the sentence

In Results there is a sentence “…it is basically possible that this mutation causes a splicing defect…” it doesn’t sound right to me (a little colloquial/unscientific). I would remove the word “basically” but it is up to you.

In Article there is no PCR conditions for reaction with Polymerase Phusion. How many cycles, temperatures for primers. Which buffer was used GC or HF. Concentrations for primers etc.?

During revision I have found also this article: https://www.ncbi.nlm.nih.gov/pmc/articles/PMC8205996/ maybe you will find it interesting (don’t worry I’m not the author of this paper you don’t need to cite it, just wanted to help)

Author Response

Reviewer #1-1

The work is highly focused on proving that mutation at the 3’ end of exon-3 is a direct or indirect cause of exon-3 skipping and thus production of d[ys]functional protein causing Desmin[e] protein aggregation and sickness of a patient. Authors did a sequencing of a big panel of genes (TrueSight Cardio NGS panel) containing more than 100 genes connected to the cardiomyopathy. My question is were there other pathogenic mutations identified (in other genes) that could be also associated with observed patient symptoms?

Answer of the authors

We thank reviewer #1 for this question. However, there were no other pathogenic mutations identified. All other identified variants can be excluded by their high minor allele frequency (MAF) as pathogenic mutations. We have incorporated this information in the revised manuscript.

Reviewer #1-2

If this was the case it should moderately weaken the overall conclusions. Patient have a point mutation in on average 50% of alleles (Illumina NGS on DNA) and based on Nanopore cDNA sequencing again ~50% (based on Fig 4) sequenced transcripts contain deletion of the whole exon-3 but only <1% sequences with the whole exon-3 had 735G>C mutation. This can be explained if we assume the expression seen in Nanopore data comes from only one allele. The <1% are “leaks” from the second allele and/or sequencing/base calling artifacts that are common for Nanopore technology (If I remember correctly 80-90% confidence in base calling of nucleotides?). So, it doesn’t convince me that there is connection between mutation and exon deletion. Are there any records of healthy people (without 735G>C mutation) with exon-3 lacking Desmine? Probably it will be hard to verify.

Answer of the authors

We have done Nanopore sequencing also with a myocardial sample from a rejected donor heart (non-failing, Figure 4B) and were unable to identify reads without DES exon-3 in this sample. Thus, it is highly likely that the missense mutation localized at the last base pair of exon-3 (determined at the genomic DNA level) causes the described splicing defect observed at the mRNA level. Please compare the coverage of the other DES exons, which is 50% higher than in exon-3 in the sample from the index patient (Figure 4C). We have identified at the genomic DNA level the mutation DES-c.735G>C by next generation sequencing and additionally by Sanger sequencing. The exon-3 skipped desmin transcript was determined by Nanopore amplicon sequencing at the mRNA level. In contrast,

Reviewer #1-3

Authors presented figure with a Western blot DES detection analysis as a merge of 2-3 Western blots. Their “control” is showing “lack” of expression of the protein with exon-3 deletion but in my opinion the lower band is present also in the control (very week signal) and because the two protein samples were not run and exposed (detected with chemiluminescence?) together it is impossible to compare those signals. Also, we can see additional bands on a membrane so the antibody is probably detecting off-target proteins or dimmers?. Please provide pictures of full western blots in supplementary data. According to supplementary data from Brodehl et al., 2012. [Dual Color Photoactivation Localization Microscopy of Cardiomyopathy-associated Desmin Mutants. 10.1074/jbc.M111.313841] Endogenus Desmin may come in two different forms ~54kDA and ~50kDA. Different isoforms are present in few cell lines. Base on that paper I’m assuming, the smaller band shouldn’t necessarily be counted as a unusual and pathogenic form?

Answer of the authors

We accept the criticism of reviewer #1. The Western blot analysis using explanted myocardial tissue from the index patient was therefore repeated and these data were incorporated into the revised manuscript. The two protein samples (patient and control) were run and exposed together. In addition, we show as suggested by the reviewer the full Western blot images in the supplementary data. For our previous manuscript (Brodehl et al. J Biol Chem, 2012), we have performed cell transfection experiments of desmin fused to an EYFP-tag. The two bands we observe in the JBC manuscript come both from the endogenous desmin and from the desmin-EYFP fusion construct and do not represent different desmin isoforms. Therefore, it is not directly comparable to this investigation using human explanted myocardial tissue. In addition, we have indicated chemiluminescence as the detection method in the revised manuscript.

Reviewer #1-4

Also (maybe it is my lack of knowledge) how often it happens that missense mutations in exons delete the whole exon (I understand that it happens with nonsense mutations but with just amino acid substitution?)?

I found this paper: Liu H.X., et al 2001. [A mechanism for exon skipping caused by nonsense or missense mutations in BRCA1 and other genes. DOI: 10.1038/83762] proving that it may happen at least in 5’ exon end but how it relates to Authors case with mutation at the 3’ of an exon. Is this paper supporting Authors findings?

I’m not fully convinced that mutation (not at the beginning of an exon) can cause whole exon deletion. Maybe the conclusion should be watered down a little to leave space for other interpretations? I don’t expect you to answers all of my questions but please do your best to convince me and future readers.

Answer of the authors

The splicing of the mRNA molecules depends on the donor and acceptor splicing sites, localized in the pre-mRNA molecules (compare Figure). These splice sites are highly conserved. According to the sequence logo the last G in the exon at the splice donor site is highly conserved. However, it is important to understand that the splicing in eukaryotic cells happens before the translation of mRNA into proteins. Therefore, the encoded amino acid sequence is not relevant.

http://www.geneinfinity.org/sp/sp_coding.html

It is known, that this donor splice site of DES exon-3 is highly sensitive for mutations. Some other groups have found splicing mutations in the intronic part of this donor splice site (see for review Brodehl A et al. Biophysical Reviews, 2018). Furthermore, our results were supported by a different manuscript where the authors describe another pathogenic DES mutation (c.734A>G) in close proximity to our described mutation also localized in the exonic part of exon-3 (Nalini A et al. 2013, Neurology India). However, the authors have not performed any functional analysis of DES-c.734A>G.

Reviewer #1-5

Very interesting work. Some small details:

Material and methods

First line: ”… right heart failure at the age of 41…” there are two spaces between “failure” and “at”.

„…diastolic dysfunction. . In addition,…” the are two dots at the end of the sentence

In Results there is a sentence “…it is basically possible that this mutation causes a splicing defect…” it doesn’t sound right to me (a little colloquial/unscientific). I would remove the word “basically” but it is up to you.

Answer of the authors

We thank reviewer #1 for recognising these errors and have corrected them in the revised version of the manuscript.

Reviewer #1-6

In Article there is no PCR conditions for reaction with Polymerase Phusion. How many cycles, temperatures for primers. Which buffer was used GC or HF. Concentrations for primers etc.?

Answer of the authors

We apologize for the missing PCR conditions and incorporate them in the revised manuscript.

Reviewer #1-7

During revision I have found also this article: https://www.ncbi.nlm.nih.gov/pmc/articles/PMC8205996/ maybe you will find it interesting (don’t worry I’m not the author of this paper you don’t need to cite it, just wanted to help)

Answer of the authors

We thank reviewer #1 for this information. However, we want to underline that it is about arrhythmogenic cardiomyopathy and not about restrictive cardiomyopathy. Therefore, we have not cited this interesting paper in our manuscript.

Reviewer 2 Report

The desmin mutation DES-c.735G>C causes severe restrictive cardiomyopathy by inducing in-frame skipping of exon-3 the authors described an interesting case of RCM treated with heart transplant

The case is very interesting and well written 

Minor points

  • The introduction is too long, please reduce it
  • Please indicate the value of RQ of CPET
  • Please indicate how days after levosimendan infusion RCH was performed 

Author Response

Reviewer #2-1

The desmin mutation DES-c.735G>C causes severe restrictive cardiomyopathy by inducing in-frame skipping of exon-3 the authors described an interesting case of RCM treated with heart transplant. The case is very interesting and well written

Answer of the authors

We thank reviewer #2 for this motivating statement.

Reviewer #2-2

Minor points: The introduction is too long, please reduce it.

Answer of the authors

According to this suggestion of reviewer #2, we shortened the introduction of the revised manuscript.

Reviewer #2-3

Please indicate the value of RQ of CPET.

Answer of the authors

We were unable to provide the respiratory quotient of cardiopulmonary exercise testing (RQ of CPET), because it was not determined.

Reviewer #2-4

Please indicate how days after levosimendan infusion RCH was performed.

Answer of the authors

We included this information in the revised manuscript.
